# Latency Reversing Agents and the Road to an HIV Cure

**DOI:** 10.3390/pathogens14030232

**Published:** 2025-02-26

**Authors:** Louis Tioka, Rafael Ceña Diez, Anders Sönnerborg, Maarten A. A. van de Klundert

**Affiliations:** 1Faculty of Medicine, Erlangen-Nürnberg, Friedrich-Alexander-Universität, 91054 Erlangen, Germany; louis.tioka@fau.de; 2Division of Infectious Diseases, Department of Medicine Huddinge, Karolinska Institutet, 17177 Stockholm, Sweden; rafael.cena.diez@ki.se (R.C.D.); anders.sonnerborg@ki.se (A.S.); 3Department of Infectious Diseases, Karolinska University Hospital, 17177 Stockholm, Sweden; 4Division of Clinical Microbiology, Department of Laboratory Medicine, Karolinska Institutet, 17177 Stockholm, Sweden

**Keywords:** latency, latency reversal, latency reversing agents, HIV cure, HIV-1, kick and kill, viral reservoir

## Abstract

HIV-1 infection cannot be cured due to the presence of HIV-1 latently infected cells. These cells do not produce the virus, but they can resume virus production at any time in the absence of antiretroviral therapy. Therefore, people living with HIV (PLWH) need to take lifelong therapy. Strategies have been coined to eradicate the viral reservoir by reactivating HIV-1 latently infected cells and subsequently killing them. Various latency reversing agents (LRAs) that can reactivate HIV-1 in vitro and ex vivo have been identified. The most potent LRAs also strongly activate T cells and therefore cannot be applied in vivo. Many LRAs that reactivate HIV in the absence of general T cell activation have been identified and have been tested in clinical trials. Although some LRAs could reduce the reservoir size in clinical trials, so far, they have failed to eradicate the reservoir. More recently, immune modulators have been applied in PLWH, and the first results seem to indicate that these may reduce the reservoir and possibly improve immunological control after therapy interruption. Potentially, combinations of LRAs and immune modulators could reduce the reservoir size, and in the future, immunological control may enable PLWH to live without developing HIV-related disease in the absence of therapy.

## 1. Introduction

### 1.1. The Latent Reservoir in Natural Infection

In people living with HIV-1 (PLWH), HIV-1 replicates by infecting immune cells, the CD4+ T cells, and monocytes. A small but especially clinically important fraction of the infected cells becomes latently infected when the viral genome is integrated in the host cell, but the cell does not readily start to produce the virus. Instead, virus production can be reactivated at random timepoints, days, years, or decades later. The viral reservoir is the reason that HIV-1 infection cannot be cured, and medication must be taken lifelong to prevent patients from developing the acquired immunodeficiency syndrome (AIDS).

In treated PLWH, about 1 in every 10,000 CD4+ T cell contains HIV-1 DNA, of which 95% is not replication-competent. While the total amount of integrated HIV-1 DNA is relatively stable over time, the fraction of replication-competent HIV-1 decreases with a half-life of approximately 5 years [1]. The composition of both the replication-incompetent and -competent reservoirs are dynamic. While the size of the replication-competent reservoir, expressed as a percentage of total CD4+ T cells, decreases over time, the size of the replication-incompetent reservoir is mostly stable. The reservoir size is mainly maintained by clonal expansion [2]. As some cells are more prone to expand and some are more prone to die, both the reservoirs show an increase in T cell clones over time [2,3]. These dynamics are intrinsically related to the variable nature of latency. While some cells may never reactivate, other cells continuously have low levels of HIV-1 RNA transcription, especially early transcripts. A large part of the cells that can fully reactivate HIV-1 replication is presumably eliminated, leading to a reduced size of the replication-competent reservoir over time. In line with these notions, it has been observed that latently infected cells, especially those forming large clones, have an increased resistance to apoptosis. Typically, such cells show signs of low-level transcription; they express low amounts of early genes, but transcription is not sufficiently activated to induce the translation of viral transactivator TAT, which is needed for full transcriptional activity [4].

### 1.2. HIV-1 Reactivation in Natural Infection

The replication-competent HIV reservoir declines over time. As the replication-incompetent reservoir is relatively stable, this decay reflects the influence of reactivation and subsequent cell death due to the cytopathic effects of HIV replication. HIV reactivation is a largely stochastic process, but the odds of reactivation are heavily influenced by the activity of the cell in which the virus is integrated. HIV-1 replicates in activated CD4+ T cells in which the TCR has been stimulated. The HIV promoter, the LTR, contains binding sites for transcription factors that are active in activated T cells, such as NF-kb, AP-1, and NFAT. In line with this notion, the activation of a (memory) T cell by antigens is an efficient way to reactivate HIV ex vivo. This reactivation presumably occurs frequently in vivo, as recurrent (e.g., flu) and chronic (e.g., CMV) infections trigger the reactivation of memory T cells. Importantly, a large fraction of the latent reservoir is situated in memory T cells specific to HIV itself, and the ex vivo stimulation of T cells with HIV antigens can efficiently reactivate the majority of latently infected cells. Thus, such T cells are likely to reactivate in the presence of other cells expressing HIV antigens. Also, other, more general corelates of immune regulation and activity, such as cytokine levels, have been shown to be related to the reservoir size and activity [5].

### 1.3. HIV Latency Reversing Agents

Many LRAs act by directly or indirectly activating transcription factors that bind to the HIV LTR and activate transcription, such as NF-kB, NFAT, AP1, and Sp1 (Table 1).

**Table 1 pathogens-14-00232-t001:** Overview of different LRAs.

LRA Class	Mode of Action	Drugs and Compounds		Past and Ongoing Studies			HIV Reactivation/Decrease in Viral Reservoir in Patients
				In Vitro	Ex Vivo	In Vivo	
PKC agonists	Activation of NF-κB	Phorbol esters	Phorbol 12-myristate 13-acetate (PMA)	Reactivation of HIV-1 transcription [6]	Reactivation of HIV-1 transcription [6,7,8]		n.a.
			Prostratin	HIV-1 latency reversal [9,10]	HIV-1 latency reversal [9]		n.a.
			12-Deoxyphorbol 13-phenylacetate (DPP)	HIV-1 latency reversal [11]	HIV-1 latency reversal [12]		n.a.
		Macrocyclic lactones	Bryostatin-1 and analogues	Reactivation of HIV-1 gene expression [12,13]	Reactivation of HIV-1 gene expression [14]	Tested in the context of malignancies [15] as well as in a phase I clinical trial [16]	no/no
		Diterpenes	Ingenol and derivatives	Reactivation of HIV-1 transcription [17,18]	Reactivation of HIV-1 transcription [17,18,19]	Tested in non-human primates [20] and patients with HIV-1 [21]	yes/yes
MAPK agonist	Activation of MAPK	Procyanidin C1		Reactivation of HIV-1 transcription [22]	Reactivation of HIV-1 transcription [22]		n.a.
Activators of Akt pathway	Activation of NF-κB	Disulfiram		Reactivation of HIV-1 transcription [23]	Reactivation of HIV-1 transcription [23,24]	Phase 1 and 2 clinical studies [25,26]	yes/no
		Hexamethylene bisacetamide (HMBA)		Reactivation of HIV-1 transcription [27]	Disruption of HIV-1 latency [28]	Phase 2 clinical study in the context of malignancies [29]	n.a.
		57704		Reactivation of HIV-1 transcription [30]	Reactivation of HIV-1 transcription [30]		n.a.
Immune modulatory LRAs	Activation of NF-κB	Toll-Like receptor agonists	TLR1/2 (Pam3CSK4)	Reactivation of HIV-1 gene expression [31]	Reactivation of HIV-1 gene expression [32]		n.a.
			TLR5 (Flagellin)		Reactivation of HIV-1 gene expression [33]		n.a.
			TLR7 (GS-9620)	Reactivation of HIV-1 gene expression [31]	Reactivation of HIV-1 gene expression [34]	Tested in non-human primates [35] Phase 1 clinical trial [36]	yes/n.a.
			TLR9 (MGN 1703)		Reactivation of HIV-1 gene expression [37]	Tested in phase 1 and 2 clinical trial [38]	yes/n.a.
	Activation of STAT5	IL agonists	IL-15 (ALT-803)	HIV-1 latency reversal [38]	HIV-1 latency reversal [39]	Tested in non-human primates [40,41] Phase 1 clinical trial [42]	n.a.
	Anti-anergic	Immune checkpoint inhibitors	anti-PD-1		HIV-1 latency reversal [43,44]	Phase 1 clinical trial in HIV-1 infected adults [45,46,47,48]	yes/no
			anti-CTLA-4			Phase 1 clinical trial in HIV-1 infected adults [45,46,47]	yes/no
Epigenetic modifiers	Opening of chromatin to allow transcription	HDACi	Trichostatin A	Reactivation of HIV-1 transcription [49]	HIV-1 latency reversal [50,51]	Extensively studied in clinical trials [52,53,54,55]	yes/no
			Trapoxin				
			Romidepsin				
			Vorinostat				
			Entinostat				
			Valproicacid				
			Fimepinostat				
			Chidamide				
			Panobinostat				
		HMTi	Chaetocin		Reactivation of HIV-1 gene expression [56,57,58]		n.a.
			BIX-01294				
		Induction of crotonylation	No agents published (ACSS2-driven latency reversal)	Reactivation of HIV-1 transcription [59]	HIV-1 latency reversal [59]		n.a.
		DNMTi	5-aza-cytidine	Reactivation of HIV-1 gene expression only in combination with other LRAs [60,61,62]	Reactivation of HIV-1 gene expression only in combination with other LRAs [60,61,62]		n.a.
			5-aza-deoxycytidine				
			zebularine				
SMAC mimetics	Activation of noncanonical NF-κB pathway	SBI-0637142		HIV-1 latency reversal [63]	HIV-1 latency reversal [63]		n.a.
		SBI-0953294 (Ciapavir)				Reactivation of HIV-1 gene expression in a humanized mouse model [64]	n.a.
		AZD5582				Reactivation of HIV-1 gene expression humanized mice and rhesus macaque studies [65]	n.a.
STAT5 sumoylation inhibitors	Activation of STAT5	1-hydroxybenzotriazol and derivatives		Induction of HIV-1 gene expression [66]	Induction of HIV-1 gene expression	Phase 1 trial [66]	n.a.
BET inhibitors	Release of P-TEFb	JQ1		HIV-1 latency reversal [18,67,68,69]	HIV-1 latency reversal [18,67]		n.a.
		MMQO					
		RVX-208					
		PFI-1					
		I-BET and I-BET151				Reactivation of HIV-1 gene expression in humanized mouse model [70]	n.a.
CCR5 antagonists	Activation of NF-κB	Maraviroc HIV-1 latency reversal [71,72]		HIV-1 latency reversal [71,72]	No consistent reactivation pattern [72]	Phase 2 clinical trial [73]	no/no
Tat vaccines	Activation of HIV-1 LTR	Tat-R5M4 protein Reactivation of latently infected cells [74]		Reactivation of latently infected cells [74]	HIV-1 latency reversal [74]	Tested for side effects and immunogenicity in mice [74]	n.a.

Many of these act on proteins downstream of the TCR and activate (parts of) the TCR stimulation-associated signalling pathways (Figure 1). Several classes of LRAs can be distinguished, such as PKC agonists, MAPK and AKT activators, and SMAC mimetics. Another major class of LRAs are chromatin modifiers, which lead to the decondensation of chromatin, allowing transcription factors to bind to the HIV LTR and activate transcription. A third class of LRAs are immune modulators, which act on cell surface receptors or pathogen recognition receptors (PRRs) to induce an activated state. Finally, various compounds and strategies to mimic, initiate, or enhance the effect of the viral transactivator TAT on HIV RNA transcription have been developed. All LRA classes are reviewed in detail below.

**Figure 1 pathogens-14-00232-f001:**
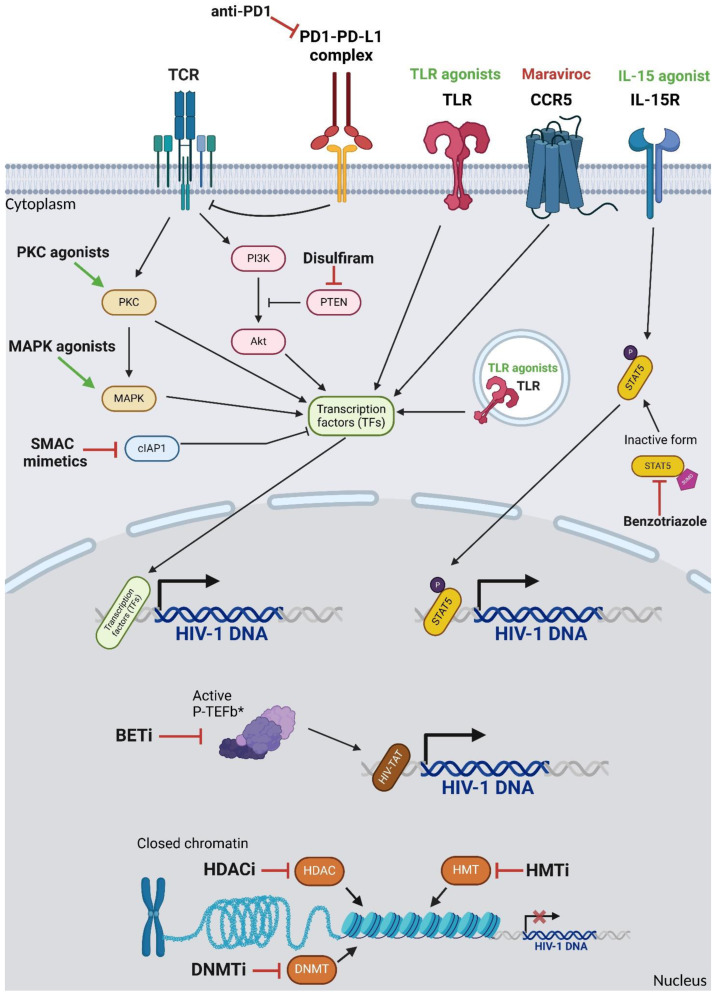
Activating protein-1. BETi: bromodomain and extra-terminal domain inhibitor. CCR5: C-C chemokine receptor type 5. cIAP1: Cellular inhibitor of apoptosis protein-1. DNA: deoxyribonucleic acid. DNMT: DNA methyltransferase. HDAC: histone deacetylase. HIV: human immunodeficiency virus. HMT: histone methyltransferase. IL-15R: interleukin 15 receptor. MAPK: mitogen-activated protein kinase. NFAT: nuclear factor of activated T cells. NF-κB: nuclear factor kappa B. P: phosphorylated. PD1: programmed cell death protein 1. PD-L1: programmed cell death 1 ligand 1. PI3K: phosphatidylinositol-3 kinase. PKC: protein kinase C. P-TEFb: positive transcription elongation factor b. PTEN: phosphatase and tensin homologue. SMAC: second mitochondria derived activator of caspases. STAT5: signal transducer and activator of transcription 5. SUMO: Sumoylated. TAT: transactivator protein. TCR: T cell receptor. TF: transcription factor. TLR: toll-like receptor. * Active P-TEFb consists of CDK9, CycT1, and Brd4 (BETi blocks Brd4 to increase viral transcription). Green arrows indicate the activation of targets by respective drug classes. Red arrows indicate the inhibition of drug targets.

## 2. Overview of Different LRAs

### 2.1. Compounds Activating Transcription Factors

#### 2.1.1. Protein Kinase C Agonists

The protein kinase C (PKC) family of kinases are serine/threonine kinases that mediate many cytoplasmatic effects related to TCR stimulation. There are nine different PKC isoforms categorized into three groups: conventional (PKCα, PKCβ, and PKCγ), novel (PKCδ, PKCε, PKCη, and PKCθ), and atypical (PKCζ, and PKCι) [75]. Conventional PKCs require both Ca^2+^ and diacylglycerol (DAG) for activation, while novel PKCs are Ca^2+^-independent, and atypical PKCs are independent of both Ca^2+^ and DAG [76]. Upon TCR stimulation, PLCγ hydrolyses membrane-associated phosphatidylinositol-4,5-bisphosphate into inositol-1,4,5-trisphosphate (IP_3_), which is released into the cytoplasm, leading to the release of Ca^2+^ from the mitochondria [77] and into DAG, which remains embedded in the membrane. DAG is a strong allosteric activator of PKC. IP3 mediates the release of Ca^2+^, another allosteric activator of PKC. LRAs that activate PKC all bind to the DAG-interacting pocket, mimicking DAG binding and initiating the downstream signalling cascades. Therefore, only conventional and novel PKCs are activated by these LRAs. PKCs inactivate the inhibitor of kB (IκB), allowing NF-κB to translocate to the nucleus and activate NF-kB responsive promoters, such as the HIV LTR (Figure 1).

PKC agonists can be grouped into three categories based on their chemical structure: phorbol esters (PMA, prostratin, and DPP), macrocyclic lactones (bryostatin-1 and analogues), and diterpenes (ingenol and derivatives). Among the phorbol esters, PMA and prostratin have been extensively studied for their ability to reactivate HIV-1 transcription. Several studies have shown that PMA and prostratin can strongly reverse HIV-1 latency when combined with ionomycin, a calcium ionophore that increases intracellular Ca2+ concentration, both in various in vitro cell models and in primary CD4+ T cells extracted from HIV-1 patients (ex vivo) [6,7,8]. However, PMA is not considered a safe candidate for clinical trials due to its tumour-promoting properties [78]. In contrast, prostratin and DPP are non-tumour promoting compounds that have been demonstrated to induce HIV-1 proviral transcription in a range of in vitro models [10,11] and to reverse latency in patient cells ex vivo [9,12]. Nevertheless, neither prostratin nor DPP have been tested in clinical trials due to concerns about toxicity. Importantly, the combination of PMA and ionomycin is probably the strongest biochemical T cell activator and LRA and has many applications in in vitro HIV (reservoir) research.

Bryostatin-1, a macrocyclic lactone PKC agonist, does not have the oncogenic effects of PMA and has been evaluated as an anticancer drug in several clinical trials [15]. In in vitro models, such as Jurkat-LAT-GFP cells, bryostatin-1 has a modest effect on HIV-1 reactivation. One clinical trial assessed the safety and efficacy of bryostatin-1 for HIV-1 eradication [16]. In this trial, bryostatin-1 did not affect PKC activity or the transcription of latent HIV-1.

Of the diterpene compounds, ingenol-3-angelate and ingenol-B have been studied in some detail. Ingenol-3-angelate is an FDA-approved drug for the topical treatment of actinic keratosis [79] and has been demonstrated to effectively reverse HIV-1 latency in vitro and ex vivo [17,80]. A recent study demonstrated that the topical administration of ingenol-3-angelate led to an overall increase in the level of HIV-1 transcription initiation, elongation, and complete transcription in skin biopsies of PLWH with actinic keratosis [21]. Ingenol-B has also been shown to be highly effective in reactivating HIV-1 in vitro as well as in purified primary CD4^+^ T cells from ART-treated HIV-1-positive individuals ex vivo [81]. In macaques, treatment with Ingenol-B and the HDACi vorinostat led to an increased viral load in the cerebrospinal fluid in one of the two non-human primates [20].

#### 2.1.2. MAPK Agonists

The activation of MAPK/ERK signalling can induce HIV-1 reactivation by signalling through AP1 and NF-κB [82,83] (Figure 1). Procyanidin C1 trimer (PC1), a plant-derived flavonoid ERK activator, has been shown to reactivate HIV-1 in both Jurkat cell line models and primary CD4+ T cells [22]. Combining PC1 with other LRAs activates multiple cellular transcription factors through distinct pathways, leading to the synergistic reactivation of HIV-1 [22].

#### 2.1.3. Activators of the Akt Pathway

Following TCR stimulation, PI3K is recruited to the plasma membrane, where it catalyzes the conversion of membrane phosphoinositide 4,5-biphosphate (PIP_2_) to phosphoinositide 3,4,5-triphosphate (PIP_3_) [84]. This results in the recruitment and subsequent activation of AKT kinase to the membrane [85]. The activation of the PI3K-Akt pathway leads to the activation and translocation of NF-κB to the nucleus (Figure 1).

The phosphatase and tensin homologue (PTEN), a lipid phosphatase, acts as the key negative regulator of the PI3K-Akt pathway by dephosphorylating Akt [86].

Disulfiram depletes PTEN, leading to the activation of AKT signalling and the reactivation of HIV in both in vitro and ex vivo cell models [23,24]. Disulfiram is an FDA-approved drug for treating alcohol use disorders and has also been tested in two clinical trials, but did not significantly reduce the size of the latent HIV-1 reservoir in patients on antiretroviral therapy (ART) [25]. A more recent phase II study showed that short-term disulfiram administration in PLWH on suppressive ART increased unspliced HIV-1 RNA at all doses, indicating that the drug may indeed activate latent HIV in vivo, but not to the levels required to significantly reduce the latent reservoir [26].

Hexamethylene bisacetamide (HMBA) transiently activates the PI3K-Akt pathway, leading to the activation of positive transcription elongation factor b (P-TEFb), which is recruited to the HIV-1-LTR, where it can initiate viral RNA transcription and viral reactivation [87]. HMBA can enhance the prostratin-induced phosphorylation and degradation of IκBα and enhance NF-κB translocation into the nucleus, thereby facilitating the transcription initiation [27].

The compound 1,2,9,10-tetramethoxy-7H-dibenzo[de,g]quinolin-7-one, also known as 57704, was identified in a compound screen to reactivate latent HIV-1 in various cell line models and CD4+ T cells from HIV-1-infected individuals on suppressive cART by agonizing the PI3K pathway [30]. The authors suggest that 57704 could serve as a scaffold for developing more potent HIV-1 latency-reversing agents, warranting further investigation [30].

#### 2.1.4. SMAC Mimetics

Second mitochondria-derived activator of caspase (SMAC) mimetics antagonize the inhibitor of apoptosis proteins (IAPs) [88]. IAPs, such as XIAP, cIAP1, and cIAP2, inhibit apoptosis by blocking caspases and SMAC mimetics bind to these IAPs, promoting their degradation [89]. In the context of HIV-1 latency reversal, cIAP1 plays a key role by constitutively degrading NF-κB-inducing kinase (NIK), thereby inhibiting non-canonical NF-κB activation [90] (Figure 1). The inhibition of cIAP1 by SMAC mimetics therefore leads to the accumulation of NIK, phosphorylation of IKKα, non-canonical activation of NF-κB, and, ultimately, reactivation of latent HIV-1 [63].

Several SMAC mimetics have been developed and evaluated in the context of a HIV-1 cure. Treatment with the SMAC mimetic SBI-0637142 enhances HIV-1 transcription and reverse latency in a JLat model system and demonstrated synergy with the HDACi panobinostat [63]). Later, Pache et al. developed another SMAC mimetic, Ciapavir (SBI-0953294), which was specifically optimized for HIV-1 latency reversal and demonstrated efficacy as a LRA [64]. Ciapavir activated HIV-1 reservoirs in a humanized mouse model without the activation of T cells [64]. Another group recently reported the activation of the non-canonical NF-κB pathway by another SMAC mimetic (AZD5582) inducing HIV-1 and SIV RNA expression in ART-suppressed humanized mice and rhesus macaques, respectively [65]. The combination of AZD5582 with a bispecific CD3 and HIV envelope molecule, designed to redirect CD8+ T cells toward HIV-infected cells expressing the envelope protein for both shock and kill effects, proved insufficient in reducing SHIV RNA levels [91]. Furthermore, a study investigated the effect of combining AZD5582 with the IL-15 super-agonist N-803 in SIV-infected, ART-suppressed rhesus macaques, demonstrating enhanced effects on latency reversal [92]. Taken together, these findings suggest that SMAC mimetics promote HIV-1 latency reversal, highlighting their potential as therapeutic agents for HIV-1 curing strategies; however, no clinical trials have yet been conducted. Interestingly, SMAC mimetics also selectively drive HIV replicating cells into apoptosis, and this effect is enhanced in vitro by combining SMAC mimetics with a PKC activator [93].

### 2.2. Epigenetic Modifiers

After integration, the HIV-1 provirus is embedded in the host cell chromatin, and the transcriptional activity of the provirus to a large extent depends on whether the chromatin is in a transcription-promoting relaxed (euchromatin) state or in a transcription-suppressing condensed (heterochromatin) state. This state is mostly regulated by modifications, such as acetylation and methylation, of the histone proteins to which the cellular DNA is bound. Epigenetic modifiers that regulate the chromatin state influence the transcription of both cellular as well as HIV-1 RNA. Compounds that inhibit epigenetic modifiers that maintain a condensed, transcriptionally silent state can allow the cellular transcription machinery to access the LTR and initiate viral RNA transcription. Among these epigenetic-based LRAs, histone deacetylase inhibitors (HDACi), histone methyltransferase inhibitors (HMTi), and DNA methyltransferase inhibitors (DNMTi) are the most widely studied.

Van Lint and colleagues were the first to demonstrate that inhibiting HDACs with HDACi (trapoxin and trichostatin A) leads to the histone hyperacetylation and activation of HIV-1 transcription, independent of NF-κB activation [49] (Figure 1). Since then, numerous HDACi have been explored as potential LRAs [52,53,54,55,94,95]. A great benefit of HDACi is that many have received FDA approval for cancer treatment, facilitating their application in HIV-1 clinical trials. Consequently, many clinical trials investigating HDACi in the context of HIV-1 have been performed [52,53,54,55]. For instance, in a phase 1/2 clinical trial involving 15 aviraemic PLWH on cART, oral panobinostat resulted in an enlargement of unspliced HIV-1 RNA and the induction of plasma viremia but did not achieve a cohort-wide reduction in total or integrated HIV-1 DNA [54]. Another proof-of-concept phase 1b/2a trial showed that HDACi romidepsin, administered to six aviraemic HIV-1-infected individuals on ART, increased HIV-1 transcription and plasma HIV-1 RNA levels without impairing T cell function or cytokine production [52]. Posttranslational histone methylation marks are crucial in establishing and maintaining a heterochromatin state, and, in line with this, inhibiting the enzymes responsible for histone methylation, histone methyl transferases (HMTs), can reactivate latent HIV-1 [58,96] (Figure 1). Although HMTis have not been studied as extensively as HDACis, several compounds targeting the enzymes responsible for histone methylation have yielded promising results ex vivo. Bouchat et al. demonstrated that two specific HMTis, chaetocin and BIX-01294 (which inhibit Suv39H1 and G9a, respectively), effectively reactivated HIV-1 expression in resting CD4+ (rCD4+) T cell cultures isolated from HIV-1-infected individuals on HAART [56,60]. Other studies have also reported that chaetocin induces latent HIV-1 expression while exhibiting minimal toxicity and without triggering T cell activation [57]. In recent years, histone crotonylation has been described as another epigenetic mechanism capable of reactivating latent HIV-1. Jiang et al. demonstrated that inducing histone crotonylation by the expression of the acyl-CoA synthetase short-chain family member 2 (ACSS2) can reactivate latent HIV-1 [59]. Moreover, they could confirm synergistic effects in combination with the PKCa PEP005 and the HDACi vorinostat, both in vitro and ex vivo settings [59]. More research has to be elicited to investigate whether LRAs that induce crotonylation or suppress decrotonylation are effective in reversing HIV-1 latency.

In addition to posttranslational histone modifications, the direct methylation of CpG residues in integrated HIV DNA can silence HIV-1 RNA transcription in a reversible manner. HIV infection can alter the DNA methylation pattern in T cells [97]. Research has demonstrated that two CpG islands present in the HIV-1 transcription start site are methylated in latently infected Jurkat cells and primary CD4+ T cells [62], thereby maintaining a transcriptionally silent and latent proviral state (Figure 1). To target the enzymes responsible for DNA methylation, several compounds have been developed that are already FDA-approved for cancer treatment. Studies utilizing these compounds in vitro and ex vivo have highlighted their potential as LRAs for reactivating HIV-1 [60,61,62]. Interestingly, the methylation of HIV DNA in treated people with HIV infection is rare, suggesting that DNA methylation does not play a direct role in HIV latency and that the inhibition of DNMTs may reactivate HIV by an indirect mechanism [98].

### 2.3. Immune Modulators

The reactivation of HIV is intrinsically related to the activation of T cells. While TCR stimulation is the prime means to activate T cells, the recognition of pathogens by other means than the TCR can lead to T cell activation. Different cytokines and cell–cell interactions modulate T cell activity and can induce the activation of T cells directly as well as modulate the way that T cells respond to TCR stimulation. Thus, factors in the T cell environment that modulate T cell behaviour, so-called immune modulators, can induce activation-related transcription factors.

#### 2.3.1. TLR Agonists

Toll-like receptors (TLRs) are pattern recognition receptors (PRRs) that are activated by specific pathogen-associated molecular patterns (PAMPs). Upon activation, TLRs initiate signalling pathways that lead to the production of pro-inflammatory cytokines that can activate T cells. Thus, their mode of action is often more indirect compared to other latency-reversing agents (LRAs).

TLR-7 agonists, such as GS-986 and GS-9620, have been extensively studied. In 2017, GS-9620 was shown to reactivate HIV-1 in peripheral blood mononuclear cells (PBMCs) from PLWH on ART, leading to increased virus production [34]. More recent research has demonstrated that both GS-986 and GS-9620 can induce viral reactivation and reduce viral reservoirs in simian immunodeficiency virus (SIV)-infected rhesus macaques on ART [35]. However, in a phase I clinical trial, Vesatolimod (GS-9620) did not reduce the latent HIV-1 reservoir in 48 HIV-1-infected individuals on cART, despite evidence of cytokine release, the induction of the expression of interferon-stimulated genes (ISGs), and other signs of lymphocyte activation [36].

The TLR-1/2 agonist Pam3CSK4 has been displayed to reactivate dormant HIV-1 in central memory CD4+ T cells and rCD4+ T cells isolated from viraemic patients by inducing transcription factors NF-κB, NFAT, and AP-1 [32]. A synthetic dual TLR2/7 agonist was later characterized by Macedo and colleagues. This compound reactivates HIV-1 via TLR2-related NF-κB induction, while the TLR7 stimulation promotes viral reactivation through TNF-α secretion from monocytes and plasmacytoid dendritic cells [31].

Other TLR agonists, such as the TLR5 agonist flagellin, have been found to activate NF-κB and reactivate latent HIV-1 in CD4+ T cells, including quiescent central memory CD4+ T cells [33]. Similarly, the TLR9 agonist MGN1703 can activate HIV-1 transcription in PBMCs from aviraemic HIV-1-infected individuals on ART [37], though clinical trials with MGN1703 showed only moderate LRA effects and no significant reduction in viral reservoirs [99,100].

Despite encouraging in vitro and ex vivo findings, the clinical application of TLR agonists as standalone LRAs has not demonstrated significant effects on the reservoir size, highlighting the need for further investigation and potentially combinatorial therapeutic approaches [101].

#### 2.3.2. IL-15 Agonist

The binding of IL-15 to the IL15 receptor results in signalling through the activation of Janus Kinase (JAK) and the further downstream activation of the primarily signal transducer and activator of transcription 5 (STAT5) [102] (Figure 1). In HIV-1 infection, endogenous IL-15 has been shown to promote CD8+ T cell expansion in ART-naïve PLWH [103] and to exhibit both latency-reversing and immune-enhancing properties by reactivating latent HIV-1 RRR. Additionally, IL-15 has been found to increase the activation of CD8+ T cells and natural killer cells, leading to the increased production of the effector molecules IFN-γ and TNFα, as well as the enhanced killing capacity of target cells [104,105]. As a result, IL-15 agonists have the potential to act as both a “shocking” and “killing” agent in HIV-1 therapy. The IL-15 superagonist ALT-803, also known as N-803, has been studied for its anti-HIV-1 effects both in vitro and in vivo [38,40,41]. While ALT-803 alone did not induce viral reactivation in ART-treated macaques infected with simian immunodeficiency virus (SIV), its administration following CD8+ lymphocyte depletion led to robust and sustained viral reactivation [41]. The first clinical trial of ALT-803 in humans, conducted in patients with hematologic malignancy relapse after allogeneic hematopoietic cell transplantation, demonstrated its safety and efficacy. In this phase 1 trial involving 33 patients, the subcutaneous administration of ALT-803 resulted in prolonged serum levels compared to intravenous administration and enhanced the expansion of NK and CD8+ T cells. The treatment was well tolerated, with no treatment-related graft-versus-host reactions [42]. This study highlights ALT-803’s safety and its potential as a promising option for future clinical trials in the context of HIV-1 eradication and combinatorial therapies. It should be noted that the induction of STAT5 by IL-15 also drives the homeostatic proliferation of CD4+ memory T cells and, thus, especially in the context of the incomplete reactivation and/or lack of killing of reactivated cells, could also contribute to the outgrowth of latently infected CD4+ T cell clones [106,107].

#### 2.3.3. Immune Checkpoint Inhibitors

HIV-1 infection leads to an ongoing inflammatory environment that promotes the accumulation of so-called exhausted T cells with impaired effector function. These exhausted or anergic T cells exhibit the increased expression of immunomodulatory receptors that downregulate their activity and capacity to react to antigens, such as Programmed Cell Death Protein 1 (PD-1) and Cytotoxic T Lymphocyte-Associated Protein 4 (CTLA-4). As a result, anergic T cells display reduced proliferative capacity and increased susceptibility to apoptosis [108,109,110]. Chronic immune activation and T cell exhaustion contribute to establishing and maintaining HIV-1 latency by preventing the (re)activation of latently infected cells [111] (Figure 1). Consequently, blocking these co-inhibitory receptors offers a dual therapeutic strategy to both restore immune function and reverse HIV-1 latency. Many immune checkpoint inhibitors have been developed and approved for cancer treatment, and such compounds may be repurposed for HIV-1 curing strategies. Some studies where immune checkpoint inhibitors were applied involved PLWH, allowing scientists to assess their effect on the viral reservoir [47,112,113].

The PD-1 blockade using receptor-blocking antibodies pembrolizumab and nivolumab has been shown to enhance HIV-1 production and increase cell-associated HIV-1 RNA without significant T cell activation in ART-suppressed CD4+ T cells ex vivo [43,44]. In a phase I trial, BMS-936559, another anti-PD-1 agent, was tested in eight HIV-1-infected adults on ART and was well tolerated, showing no significant T cell activation but enhancing HIV-1-specific immunity in some participants [48]. This study paves the way for further clinical trials investigating BMS-936559 in HIV-1 latency reversal. Notably, combining different immune checkpoint inhibitors appears to be more effective in inducing HIV-1 latency reversal than using them individually. In a study by Harper et al., the CTLA-4/PD-1 blockade was shown to be more effective than PD-1 monotherapy at reversing SIV latency and reducing the levels of the integrated virus in CD4+ T cells of SIV-infected macaques on long-term ART [114]. Similarly, a phase I clinical trial demonstrated that while anti-PD-1 alone did not affect HIV-1 latency or the size and composition of the latent reservoir, the combination of anti-PD-1 and anti-CTLA-4 modestly increased HIV-1 RNA levels [47].

#### 2.3.4. CCR5 Antagonist

CCR5 is a critical co-receptor that is essential for HIV-1 entry into host cells. Maraviroc (MVC), a small-molecule CCR5 antagonist approved by the FDA, has been developed to inhibit HIV-1 entry by blocking CCR5 (Figure 1). It has shown promise in vitro as well as in clinical trials for managing R5-tropic HIV-1 infections [115,116] and is currently being investigated for its potential role in HIV-1 latency reversal, thus offering a dual therapeutic strategy to address both active and latent infections.

MVC has been found to increase levels of unspliced viral RNA and is associated with the enhanced expression of NF-κB-dependent genes in rCD4+ T cells from HIV-infected individuals on suppressive ART [117]. López-Huertas and colleagues demonstrated that MVC may serve as a novel potential LRA since it reverses HIV-1 latency in in vitro HIV-1-latency. They found that MVC alone was as efficient as when used in combination with bryostatin-1 [71]. This antagonistic effect was likely due to bryostatin-1’s ability to decrease CCR5 expression levels [71]. A recent study showed that MVC can induce HIV-1 expression ex vivo in CD4+ T cells from patients, but the response is variable and patient-specific [72]. While significant activation was observed at high concentrations of MVC in some models, no consistent patterns emerged, indicating the need for further research to clarify its clinical relevance [72].

#### 2.3.5. Benzotriazole Derivatives

A chemical drug screening of a natural product library in a model of central memory T cells (TCM) latently infected cells identified as 1-hydroxybenzotriazole (HOBt) and its derivatives as effective LRAs. These compounds, which currently have no known biological function, were shown to reactivate latent HIV-1 in multiple cellular models, including ex vivo cultures from aviremic patients, without causing cellular activation, proliferation, or toxicity [118]. Mechanistically, the authors suggested that benzotriazole derivatives block the SUMOylation of phosphorylated STAT5, promoting STAT5 binding to the HIV-1 LTR and thereby increasing its transcriptional activity [118] (Figure 1). A follow-up study revealed that the compounds could reactivate HIV-1 in a primary cell latency model and augment interleukin-15’s ability to stimulate latent HIV-1 reactivation ex vivo [66]. Additionally, this family of compounds was found to promote immune effector functions in vitro without toxicity or global immune activation, and preliminary studies in mice indicated a lack of acute toxicity [66]. The constitutive activation of STAT5 signalling may increase the basal homeostatic proliferation of CD4+ T cells [107]. Thus, despite initial HIV-1 latency reversal, the activation of STAT5 by benzotriazole derivatives may lead to the expansion of certain subsets within the HIV reservoir.

### 2.4. TAT Mimics and Enhancers

#### 2.4.1. Tat Vaccines

The HIV-1 Tat protein (Transactivator of Transcription) plays a pivotal role in the viral life cycle and pathogenesis. Tat is essential for the transcriptional activation of the HIV-1 genome, promoting the production of viral RNA and proteins by interacting with the host cell’s transcription machinery. This interaction involves binding to the transactivation response (TAR) element present in the HIV-1 LTR [119]. Given the specificity and efficiency of Tat in activating HIV-1 transcription, along with the established safety of the bioactive recombinant Tat protein in various clinical vaccine trials [120,121], generating a recombinant Tat protein as a new HIV-1 latency activator appears feasible. The Zhang group successfully produced an attenuated HIV-1 Tat protein named Tat-R5M4 with reduced cytotoxicity and immunogenicity while preserving its potent transactivation and membrane-penetration capabilities, utilizing a mutation accumulation strategy [74]. They further demonstrated that Tat-R5M4, combined with HDACi, activated HIV-1 in rCD4^+^ T lymphocytes from HIV-1-infected individuals undergoing suppressive cART [74]. In vivo, studies in wild-type mice confirmed that Tat-R5M4 is safe and does not adversely affect the physiological function of major organs [74]. However, it must be taken in account that Tat can penetrate the central nervous system (CNS) and interact with microglia, astrocytes, and even neurons. The presence of extracellular Tat has been shown to cause dysregulated gene expression, chronic cell activation, inflammation, neurotoxicity, and structural brain damage [122,123,124]. Thus, overexpressing Tat may lead to neuronal damage and favour the development of HIV-1-associated neurocognitive disorders (HANDs). Nevertheless, mutated Tat proteins could represent a promising strategy for novel LRAs in HIV-1 curing approaches. A large drawback is the delivery, as TAT by itself does not pass through cell membranes.

#### 2.4.2. BET Inhibitors

In the early phase of HIV infection and in cells expressing only low levels of (early) HIV genes, RNA polymerase II (RNA Pol II) is often paused during early elongation [125]. For transcription to proceed productively, the viral transactivator of the transcription (TAT) protein needs to be translated. Subsequently, TAT binds to the TAR element in the 5’ stem loop of the HIV RNA [126]. This interaction recruits super-elongation complexes, including p-TEFb (positive transcription elongation factor b), which is composed of CyclinT1 and CDK9. The positive transcription elongation factor b (p-TEFb) positively regulates HIV RNA elongation by antagonizing negative elongation factors and phosphorylating RNA Pol II’s C-terminal domain, promoting both the elongation and co-transcriptional processing of nascent RNA [127,128]. On the HIV-1 LTR, the bromodomain and extraterminal (BET) protein Brd4 interacts with p-TEFb and blocks the binding of TAT, thus hindering transcriptional elongation [129] (Figure 1). In recent years, small-molecule BET inhibitors, represented by JQ1, have emerged and have been evidenced to antagonize BRD4’s inhibitory effect by occupying its bromodomains, thereby reactivating HIV-1 transcription [69].

Another study found that the BET inhibitors RVX-208 and PFI-1 can reactivate HIV-1 in latently infected Jurkat and in ex vivo rCD4+ T cells from cART-treated patients [67]. Notably, both compounds have already been investigated in the contexts of cardiovascular diseases and oncology, respectively [130,131].

## 3. Combinations of LRAs

Several combinations of LRAs have been tested, and the results indicate that such combinations may be more efficient than treatment with single agents. In particular, studies that combine LRAs with different modes of action are of interest. For instance, HDACi (entinostat) combined with the protein kinase C agonist bryostatin-1 produced more significant HIV-1 latency reversal and viral outgrowth than different HDACi alone [51]. Also, combination treatments involving one HMTI and either a HDACi (suberoylanilide hydroxamic acid) or prostratin exhibited a greater reactivation potential than the compounds alone [56], and the combination of AZD5582 with the IL-15 super-agonist N-803 was shown to have enhanced effects on latency reversal in a SHIV-infected macaque model [92].

In particular, combinations of transcription-initiating/inducing agents and transcription progression-inducing BET inhibitors were found to synergistically activate HIV-1 expression in in vitro HIV-1 latency models of T-lymphoid and myeloid lineages [18]. Comparable results were observed when using bryostatin-1/ingenol-B and JQ1 combinations in ex vivo cultures of rCD4+ T cells isolated from PLWH on cART [18]. Another quinoline-based BETi, MMQO (8-methoxy-6-methylquinolin-4-ol), reactivates viral transcription in JLat cells and shows enhanced potency when combined with other LRAs [68]. In a phase 1 clinical trial, it has been demonstrated that the administration of panobinostat and interferon-α2a could selectively kill a significant part of the non-clonal intact proviruses, resulting in a measurable alteration in the reservoir composition [132].

## 4. Conclusions and Discussion

A large shortcoming of this review, and as such much work on HIV-1 latency, is the strong focus on T cells, despite it being well known that other cells, such as macrophages, also support HIV-1 replication. It has also been demonstrated in numerous studies that macrophages can harbour latent HIV-1 [133,134]. Some isolated HIV-1 curing cases have been described in patients that received a delta-32 stem cell transplant. In these cases, the patient T cells—but not macrophages—were eradicated and replaced by cells that lack the CCR5 HIV-1 coreceptor and therefore are not permissive to HIV replication. Such cases demonstrate that the amount of HIV-1 produced by macrophages is presumably low and does not extensively spread between macrophages in the absence of CD4+ T cells. Nevertheless, the amount of HIV-1 produced by (latently) infected macrophages may be large enough to re-initiate HIV-1 replication after an effective shock-and-kill therapy of the CD4+’T cell population. Very little is known about HIV-1 replication in macrophages in patients with suppressed viraemia. It has been demonstrated that microglial cells, which are brain resident macrophages, can harbour replication-competent, reactivatable HIV-1 [135]. The dynamics of such reservoirs should be taken into account when evaluating the potential of LRAs—for instance, in regard to whether or not they can cross the blood–brain barrier [135].

Although the application of LRAs in vivo has not led to significant impacts on the viral reservoir, several studies have demonstrated that LRAs can reactivate viral RNA transcription in PLWH. Notably, while LRAs reactivate HIV-1 replication, they may not induce the full-blown HIV-1 replication that kills the reactivated cells by the cytopathic effects, especially since at least a fraction of the latent reservoir has increased resistance to apoptosis. Thus, it seems increasingly clear that besides a ‘kick’ by an LRA, an efficient strategy to induce the ‘killing’ is of vital importance in strategies that aim to reduce or purge the viral reservoir in PLWH. Notably, various strategies to increase the immune response to HIV-1 in PLWH have been developed, and many of these have been tested alone (e.g., anti-PD-1 antibodies) and have been approved for use in patients, such as neutralizing antibodies. It has been noticed that neutralizing antibodies potentiate antiviral immune responses, possibly by efficiently delivering viral antigens to antigen presenting cells. Recent results seem to indicate that this can lead to reduced and delayed viral rebound, but that combinations of broadly neutralizing antibodies with other immune and/or reservoir modulating agents, such as LRAs, may be required to induce and sustain control of the infection [136].

In the near future, clinical trials testing combinations of LRAs and immune stimulating agents will reveal if such strategies can efficiently decrease or even purge the reservoir. Notably, some parts of the reservoir may be easier to purge than others. For instance, the part of the reservoir residing in HIV-specific T cells may be more vulnerable to strategies that activate antiviral responses concomitant with more generally activating LRAs. Importantly, strategies that reduce the reservoir may also be of great benefit to patients as they may reduce the inflammation associated with the continuous reactivation of latently infected cells.

## Data Availability

No new data were created or analyzed in this study. Data sharing is not applicable to this article. A BioRender file of the figure is available on request.

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
