# Peer review of "Latency Reversing Agents and the Road to an HIV Cure"

_pathogens, 2025, doi:10.3390/pathogens14030232_

Round 1
Reviewer 1 Report
Comments and Suggestions for Authors
In the review “Latency reversing agents and the road to HIV Cure” Toika and colleagues summarize in vitro, ex vivo, animal studies and clinical studies that have been undertaken to evaluate the potential for latency reversing agents to purge the latent HIV reservoir and ultimately clear HIV from an infected individual.
Overall, the manuscript is well written and would be of interest to readers of Pathogens. I do have some general comments which I think could improve the manuscript.
1. 1. The manuscript is divided into sections based on the different classes of LRAs and there is one summary figure to describe their mechanism of action. However, I cannot find any instances where this figure is cited. It would be beneficial when discussing the mechanism of different LRAs in the text to refer to this figure and provide the reader with a visual aid to understanding the text.
2. 2. I think what is lost a bit in the manuscript is the clinical efficacy of the different tested LRAs. In the abstract the authors say that “immune modulators have been applied in PLWH and were shown to decrease the viral reservoir and/or improve immunological control after therapy interruption.” I could not find this information in the text and cannot recall that any of the clinical trials demonstrated a decrease in the viral reservoir. I may have missed this but I believe it would be extremely helpful to summarize the efficacy from clinical trials in the Table by first separating clinical from animal studies in the in vivo column and adding a column next to the clinical part for decrease in viral reservoir yes or no. Perhaps an additional column for reactivation of HIV yes or no.
3. 3. Line 454 to 458 It is not at all clear to me how the HIV cure in the berlin patient demonstrates that HIV replication from macrophages is low but large enough to re-initiate HIV replication????
4. 4. There are some typos/ confusing sentences: line 51 – decades, line 55 missing + on CD4, 132 space missing after period, line 133 binging, lines 232 to 233 missing a period and maintaining, line 255 HMTis, line 301 should this be aviraemic ? line 316-317 act as both as a .. line 370 HIV-1 lantency in in in vitro HIV latency ??
Author Response
Response to reviewer 1:
Dear reviewer, thank you for your comments, we highly appreciate your comments and have implemented the suggested improvements in the manuscript text. Please find the point-by-point answers to your comments below. In total 16 references have been added, please note that added references have not been tracked with ‘track changes’
- The manuscript is divided into sections based on the different classes of LRAs and there is one summary figure to describe their mechanism of action. However, I cannot find any instances where this figure is cited. It would be beneficial when discussing the mechanism of different LRAs in the text to refer to this figure and provide the reader with a visual aid to understanding the text.
Response: We agree to this important point and have added citations referring to the figure to sentences in the manuscript where appropriate to aid the reader, at lines: 144, 181, 191, 219, 256, 269, 289, 338, 369, 393, 417 and 468.
- I think what is lost a bit in the manuscript is the clinical efficacy of the different tested LRAs. In the abstract the authors say that “immune modulators have been applied in PLWH and were shown to decrease the viral reservoir and/or improve immunological control after therapy interruption.” I could not find this information in the text and cannot recall that any of the clinical trials demonstrated a decrease in the viral reservoir. I may have missed this but I believe it would be extremely helpful to summarize the efficacy from clinical trials in the Table by first separating clinical from animal studies in the in vivo column and adding a column next to the clinical part for decrease in viral reservoir yes or no. Perhaps an additional column for reactivation of HIV yes or no.
Response: It is indeed true that most clinical trials have not shown any reduction in the viral reservoir, although some reactivation and alteration of the reservoir has been observed. To be more clear on this from the beginning, we have adapted the abstract text (lines 19-20). To provide a clearer overview of the HIV reactivating and reservoir reducing properties demonstrated in clinical trials, we added an extra column to the table as the reviewer suggested. in this column it is summarised whether clinical trials led to HIV reactivation and/or a decrease in HIV reservoirs or if no such findings have been reported yet (n.a.). Also lines 593-596 have been added, describing a study in which significant alterations in the reservoir were detected in a clinical trial, indicating the eradication of at least a subset of latently infected cells.
After careful consideration, we agree that the abstract text is not sufficiently backed in the abstract (lines 21-24) manuscript and we have altered it. Briefly, the notion was based on recent exciting presentations at different conferences (e.g. HIV Nordic, CROI) showing that broadly neutralising antibodies may have a protective effect that increases immunity. Although very interesting, we consider these data insufficiently established to justify the statement in the abstract as it was, therefore we have adapted it. We have also added information and a reference to the discussion concerning the vaccine effect of BrNaB’s (lines 528-531).
- Line 454 to 458 It is not at all clear to me how the HIV cure in the berlin patient demonstrates that HIV replication from macrophages is low but large enough to re-initiate HIV replication????
Response: We agree that the text was not clear and have adapted and elaborated it to convey the information we wanted to convey (lines 502 - 509). Briefly, as no virus could be detected in the Berlin patient this indicates that at least self-amplifying or spreading infection in macrophages does not occur.
- There are some typos/ confusing sentences: line 51 – decades, line 55 missing + on CD4, 132 space missing after period, line 133 binging, lines 232 to 233 missing a period and maintaining, line 255 HMTis, line 301 should this be aviraemic ? line 316-317 act as both as a .. line 370 HIV-1 lantency in in in vitro HIV latency ??
Response: Thank you for identifying these mistakes, we have corrected all errors that you pointed out, as well as critically revised the rest of the manuscript for typos and other errors.
Reviewer 2 Report
Comments and Suggestions for Authors
In this review article, Tioka L. et al. summarized several families of LRAs, both alone and in combination. While the article is informative, it could benefit from further refinement before acceptance for publication.
- In line 36, the text states, “In treated PLWH, about 1 in every 10.000 CD4+ T cells contains HIV-1 DNA, of which 95% is not replication-competent.” Is this accurate? Or is it 10,000? Additionally, the statement “…both reservoirs show an increase in T cell clones over time…” requires clarification. Are the reservoirs stabilized, increased, or both over time? A broader review of relevant literature with more citations would strengthen this section.
- In section 1.3, the suppression of Sp1 should be mentioned as another factor involved in HIV latency, alongside NF-κB, NFAT, and AP-1.
- In Table and/or Section 2.2, consider adding a brief discussion on a new class of epigenetic LRAs that induce crotonylation or suppress decrotonylation, as recently reported.
- Regarding STAT5 activation, further exploration is needed to determine whether its activation could lead to the homeostatic proliferation of memory CD4+ T cells. If so, discuss whether this could ultimately increase, rather than decrease, HIV reservoirs, even if latency is initially or temporarily reversed.
- For the figure on page 5, evaluate whether DNMTs are genuinely involved in HIV latency in CD4+ T cells isolated from PWH on ART, despite evidence that DNMT inhibitors (DNMTi) can disrupt latent HIV in in vitro cell line models (e.g., J-Lat).
- Sections 2.1.4 and 3 discussing IAPi/SMACi could be expanded to incorporate findings from studies that show enhanced latency reversal when IAPi is used in combination with other agents.
- In section 2.4.1, further analysis is needed on whether Tat overexpression could result in neuronal damage in the central nervous system (CNS).
- In the Conclusion section, it would be valuable to include a brief discussion of emerging studies focusing on the brain microglia reservoir.
Author Response
Response to reviewer 2:
Dear reviewer, thank you for your comments, we highly appreciate your comments and have implemented the suggested improvements in the manuscript text. Please find the point-by-point answers to your comments below. In total 16 references have been added, please note that added references have not been tracked with ‘track changes’.
- In line 36, the text states, “In treated PLWH, about 1 in every 10.000 CD4+ T cells contains HIV-1 DNA, of which 95% is not replication-competent.” Is this accurate? Or is it 10,000? Additionally, the statement “…both reservoirs show an increase in T cell clones over time…” requires clarification. Are the reservoirs stabilized, increased, or both over time? A broader review of relevant literature with more citations would strengthen this section.
Response: Indeed, the number should be 1 in 10,000, we thank the reviewer for pointing this out and have corrected the manuscript correspondingly. We agree that the text about the number of HIV-1 clones could be clearer, and have rewritten and elaborated the text, and have added more references as you suggested (lines 43-47, 51-52).
- In section 1.3, the suppression of Sp1 should be mentioned as another factor involved in HIV latency, alongside NF-κB, NFAT, and AP-1.
Response: Thank you for this remark. Sp1 has been added to the transcription factors involved in HIV latency (line 77).
- In Table and/or Section 2.2, consider adding a brief discussion on a new class of epigenetic LRAs that induce crotonylation or suppress decrotonylation, as recently reported.
Response: We agree that this information may be valuable and have added crotonylation to the table, and discuss the mechanism and potential of influencing histone crotonylation in the context of HIV latency in suggestion 2.2 as suggested (lines 276-283).
- Regarding STAT5 activation, further exploration is needed to determine whether its activation could lead to the homeostatic proliferation of memory CD4+ T cells. If so, discuss whether this could ultimately increase, rather than decrease, HIV reservoirs, even if latency is initially or temporarily reversed.
Response: We have added the notion that iL-15 mediated STAT5 induction may increase homeostatic CD4+ memory T cell proliferation to section 2.3.2 (lines 356-359) and 2.3.5 (lines 422-428)
- For the figure on page 5, evaluate whether DNMTs are genuinely involved in HIV latency in CD4+ T cells isolated from PWH on ART, despite evidence that DNMT inhibitors (DNMTi) can disrupt latent HIV in in vitro cell line models (e.g., J-Lat).
Response: In line with your suggestion, we have investigated the literature and found that indeed methylation of the HIV promoter is not evident in PLWH on ART. We have added this information to the manuscript (lines 292-295), as well as information on the effect of HIV infection and treatment on host DNA methylation patterns (lines 285-286). Together, these findings suggest that DNMTi’s may disrupt latency by indirect mechanisms (lines 294-295).
- Sections 2.1.4 and 3 discussing IAPi/SMACi could be expanded to incorporate findings from studies that show enhanced latency reversal when IAPi is used in combination with other agents.
Response: Thank you for this important comment. We expanded section 2.1.4 by three studies demonstrating the effect on HIV latency when combining SMAC mimetics with other agents (lines 230-235 and 238-240), and included a reference to the enhanced effect of combining SMAC mimetic with IL-15 super agonist to section 3 (combinations of LRAs, lines 485-486).
- In section 2.4.1, further analysis is needed on whether Tat overexpression could result in neuronal damage in the central nervous system (CNS).
Response: We agree that this is an important consideration especially when it comes to testing such agents in clinical trials. We therefore included a paragraph discussing the role of Tat in the brain and its potential to cause neuronal damage in the relevant section (lines 447-452).
- In the Conclusion section, it would be valuable to include a brief discussion of emerging studies focusing on the brain microglia reservoir.
Response: A very interesting topic indeed! We have added a paragraph to describe such studies and now describe the potential impact of these findings on the evaluation of LRAs in ‘shock and kill’ strategies (lines 511-516).
Round 2
Reviewer 1 Report
Comments and Suggestions for Authors
The authors have addressed my comments.